# Effects of Polymer Blending on the Performance of a Subcutaneous Biodegradable Implant for HIV Pre-Exposure Prophylaxis (PrEP)

**DOI:** 10.3390/ijms22126529

**Published:** 2021-06-18

**Authors:** Linying Li, Christine Areson, Ariane van der Straten, Leah M. Johnson

**Affiliations:** 1Engineered Systems RTI International, Durham, NC 27709, USA; ali@rti.org (L.L.); careson@rti.org (C.A.); 2Center for AIDS Prevention Studies, Department of Medicine, University of California San Francisco, San Francisco, CA 94104, USA; arianevds@gmail.com; 3ASTRA Consulting, Kensington, CA 94708, USA

**Keywords:** poly(ε-caprolactone) (PCL), biodegradable polymer, polymer blends, HIV pre-exposure prophylaxis (PrEP), tenofovir alafenamide, subcutaneous implant, long-acting drug delivery

## Abstract

Long-acting (LA) HIV pre-exposure prophylaxis (PrEP) can mitigate challenges of adhering to daily or on-demand regimens of antiretrovirals (ARVs). We are developing a subcutaneous implant comprising polycaprolactone (PCL) for sustained delivery of ARVs for PrEP. Here we use tenofovir alafenamide (TAF) as a model drug. Previously, we demonstrated that the release rates of drugs are controlled by the implant surface area and wall thickness, and the molecular weight (MW) of PCL. Here, we further advance the implant design and tailor the release rates of TAF and the mechanical integrity of the implant through unique polymer blend formulations. In vitro release of TAF from the implant exhibited zero-order release kinetics for ~120 days. TAF release rates were readily controlled via the MW of the polymer blend, with PCL formulations of higher MW releasing the drug faster than implants with lower MW PCL. Use of polymer MW to tune drug release rates is partly explained by PCL crystallinity, as higher PCL crystalline material is often associated with a slower release rate. Moreover, results showed the ability to tailor mechanical properties via PCL blends. Blending PCL offers an effective approach for tuning the ARV release rates, implant duration, and integrity, and ultimately the biodegradation profiles of the implant.

## 1. Introduction

During the past decade, significant advances have been made in the development of biodegradable polymeric materials for biomedical applications, including surgical sutures, vascular grafts, dental repairs, and drug delivery systems [1,2,3,4,5,6,7]. The ability for these polymers to break down inside the body eliminates the need for an inconvenient and sometimes painful removal of the material. The most widely explored biodegradable polymers belong to the aliphatic polyester family, including polylactide (PLA), polyglycolide (PGA), and polycaprolactone (PCL). PCL, a semi-crystalline polyester, offers tailorable mechanical properties, exceptional biocompatibility, and good processability for shaping and manufacture [8,9,10]. PCL has extensive use in LA drug delivery systems, including a contraceptive implant [11] and a urethral implant (Urolon™) [12]. Moreover, PCL is particularly amenable to blending with other polymers [8,9,10], to produce drug delivery systems with programable biodegradation properties, mechanical integrity, and permeability. For example, Ma et al. showed an improved release rate of paclitaxel from PCL matrices that were blended with poloxamer 188, in microsphere form. The porosity of the microsphere surface increased by increasing poloxamer 188 in the PCL matrix blend, thus allowing for faster release at a controlled rate [13]. Tamboli et al. also examined a pentablock copolymer of (PLA-PCL-PEG-PCL-PLA) to control the release rate of steroids with minimal burst release [14]. The ratios of PCL and PLA were adjusted to alter the crystallinity of the copolymer, thus altering the drug release kinetics. In addition, blending homopolymers of differing molecular weights (MW) offers a simple approach for controlling the drug release from polymeric implant systems [15,16,17]. For instance, Solorio et al. evaluated the effect of blending various MWs of poly(D,L-lactic-co-glycolic acid) (PLGA) to develop in situ forming implants with customizable properties [18]. Similarly, blends of PCL homopolymers of different MW could be used to tailor the properties of drug delivery systems. However, the effect of PCL MW blends on the performance of polymeric implants remains largely unexplored.

Biodegradable drug delivery systems with tunable release kinetics can benefit strategies for LA HIV-PrEP. To address the global HIV epidemic, LA HIV-PrEP offers the potential to improve drug PK profile and bioavailability and adherence over an extended period by eliminating the burden of daily or on-demand regimens of ARV drugs. In particular, LA polymeric implants that sustainably release drugs enable discretion of use, simplified dosing regimens, and retrievability during the therapeutic duration. Several LA implants containing potent ARVs are being developed for HIV PrEP. For example, a subcutaneous implant formed from a medical-grade titanium reservoir plugged with a nanofluidic membrane that houses an array of slit-nanochannels showed sustained delivery of tenofovir alafenamide (TAF) and demonstrated partial protection from HIV transmission in rhesus macaques [19]. A reservoir implant comprised of a permeable polyurethane membrane delivered TAF for over 90 days in vitro and in vivo [20,21]. Another matrix-style implant containing the investigational drug 4’-ethynyl-2-fluoro-2’-deoxyadenosine (EFdA, islatravir), a highly potent nucleoside reverse transcriptase translocation inhibitor (NRTTI) [22,23,24,25], provided prolonged EFdA release > 6 months in preclinical studies [26]. Although PCL and PLA have been explored to generate monolithic matrices of the implant, an EFdA implant comprising non-biodegradable ethylene-vinyl acetate (EVA) has recently advanced into a phase 3 clinical trial. Intracellular concentrations of EFdA-triphosphate remained above the expected protective threshold (i.e., 0.05 pmol/10^6^ PBMC cells) in humans for 12 weeks and the implant was projected to last >1 year [24,25,26]. To the best of our knowledge, the majority of the PrEP implants comprise non-erodible polymers that require surgical removal procedures upon depletion of the drug. A vital need exists to advance LA biodegradable implants for HIV PrEP.

RTI is developing a biodegradable subcutaneous implant comprising PCL that sustainably release ARVs for HIV PrEP. We have previously demonstrated the delivery of TAF for 6 months in vitro and in vivo and we have recently expanded the function of this implant as an MPT for HIV PrEP and contraception [27,28,29,30]. The release of the drug from the reservoir-style implant is controlled by several parameters, including the surface area, wall thickness, and properties of the PCL [28]. Building on feedback from socio-behavioral research [31], this biodegradable LA implant could minimize patient–clinic interactions and eliminate the need for removal procedures at the clinic to reduce the burden on healthcare systems in low- and middle-income countries. 

Herein, we report the further advancement of this biodegradable implant by developing new PCL polymer formulations to tailor the drug release rates, implant duration, and mechanical integrity. We describe the PCL properties that affect the performance of the implants, including the MW of the polymer, as well as the quantity and size of the crystalline regions. Although these studies focus on implants containing TAF as our model ARV, the polymer formulations and designs are readily translatable to alternative drugs including other ARVs for HIV PrEP.

## 2. Results and Discussions

### 2.1. Physical and Thermal Properties of PCL Formulations and Implant Performance

#### 2.1.1. Pure Non-Blended PCL

We evaluated the in vitro performance of a reservoir-style implant that comprises a biodegradable PCL membrane encapsulating a formulation of active pharmaceutical ingredient (API) (Figure 1). Once inserted subcutaneously, biological fluid penetrates the implant and dissolves the API that is encapsulated within the reservoir of the implant. The solubilized API partitions into the PCL membrane and diffuses passively through the membrane into the subcutaneous space surrounding the implant to ultimately achieve sustained release kinetics. The release rate of the API is controlled by various parameters, including the physical dimension of the implant, API formulation, and properties of PCL [27,28]. PCL biodegrades via bulk hydrolysis through random chain scission as aqueous solution permeates through the polymer, and this biodegradation process typically requires years (e.g., 1–2 years) for complete bioresorption [32]. The faster drug release process is thus decoupled from bioerosion of the PCL, enabling zero-order release profiles of the drug from the implant. Since biodegradation rates of PCL scale with the initial MW of the polymer [8,32], we evaluated three starting materials comprising pure PCL with different weight average molecular weight (MWs): (Corbion; PC-08: MW of 51 kDa, polydispersity (PD) of 2.4; PC-12: MW of 72 kDa, PD of 2.6; PC-17: MW of 106 kDa, PD of 2.6). We will refer to these three PCL types as ‘non-blended PCL’ or ‘pure PCL’ throughout the paper.

Implants comprising extruded tubes of PC-08, PC-12, or PC-17 and containing a formulation of 2:1 TAF/sesame oil were evaluated using in vitro release assays. All PCL tubes were produced using a hot-melt, single screw extrusion process with solid PCL pellets to produce tubes with a wall thickness of 100 µm and an outer diameter (OD) of 2.5 mm. Figure 2 shows the cumulative release profiles of implants fabricated with each pure PCL type (see Appendix A for the cumulative percentage drug release profile). TAF releases at a higher rate from implants comprising PC-17 (0.36 ± 0.02 mg/day) compared to implants comprising PC-12 (0.12 ± 0.01 mg/day) and PC-08 (0.07 ± 0.01 mg/day), which showed the lowest release rates among the three PCL types. The differences in drug release rates are likely attributed to the distinct properties of PCL with different MWs. As a semi-crystalline polymer, PCL consists of both amorphous regions amenable to drug transport when above the polymer glass transition temperature (Tg), and crystalline regions which pose a transport barrier that restricts passive diffusion of drug molecules. Since PCL has a Tg of −60 °C, small molecules and fluid can readily penetrate the amorphous regions of the PCL membrane and diffuse passively through the membrane, allowing for sustained release of TAF. Thus, the MW of PCL, the quantity of crystallinity, and crystallite size could potentially affect the release rate of drugs from the implant. 

The non-blended PCL tubes were further characterized using differential scanning calorimetry (DSC), gel permeation chromatography (GPC), and X-ray diffraction (XRD). Because gamma irradiation (18–24 kGy) was used to sterilize the implants after fabrication for in vivo applications, we also evaluated the effect of gamma irradiation on the properties of pure PCL formulations. Table 1 shows the weight average molecular weight (MW) and % crystallinity of non-blended extruded tubes made from three types of pure PCL before and after gamma irradiation, as well as the approximate average release rate of TAF from implants comprising these non-blended PCL tubes. All gamma-irradiated PCL tubes exhibited a slight decrease in MW and a minimal change in the % crystallinity. DSC analysis of all the PCL extruded tubes showed a melting endotherm with a narrow peak near 60 °C (Figure 3a), the characteristic melting temperature (Tm) of PCL [33]. However, the % crystallinity for each PCL type differed: PC-08 showed the highest % crystallinity (59.69 ± 1.10) compared to PC-12 (58.2 ± 0.13) and PC-17 (53.2 ± 0.12) for tubes with a 100 µm wall thickness. The crystallite size of each sample was also calculated based on the DSC results. As shown in Table 2, three types of pure PCLs showed comparable crystallite sizes (~27 nm) and no significant changes in the crystallite size of the PCL were observed after exposure to gamma irradiation. 

We also evaluated the crystallite size of the PCL extruded tubes using DSC and XRD. For the DSC studies, the three types of pure PCL formulations showed comparable crystallite sizes (~27 nm) and no significant changes in the crystallite size of the PCL were observed after exposure to gamma irradiation (Table 2). Similar to the DSC studies, XRD analysis also shows that the crystallite sizes of PC-08, PC-12, and PC-17 were comparable, where PC-08 total crystallite size was 19.1 nm (10.7+8.4), PC-12 was 18.9 nm (10.6+8.3), and PC-17 was 21.6 nm (11.7+9.9) post sterilization (Table 2). Moreover, the extruded tubes comprising PC-08, PC-12, and PC-17 showed similar diffraction patterns (Figure 3b). All the diffractograms exhibit two intense Bragg peaks at 2θ near 21.3° and 23.7°, correlating to diffraction of the (110) and (200) planes of the PCL crystallite, respectively [34,35]. Overall, both techniques indicate a crystallite size in a similar order of magnitude for three pure PCL types, however, PCL with lower MW showed a higher degree of crystallinity. The quantity of crystalline regions is likely responsible for the differences in the drug diffusion kinetics [36,37]. For example, PC-08 exhibited the highest degree of crystallinity, posing more barriers for the diffusion of the drug, leading to the lowest release rate from the implant among three non-blended PCLs. Thus, selecting PCL materials with an appropriate MW, crystallite size, and % crystallinity is critical for achieving the desired drug release rate and duration of release for the implant. By testing PCL with different starting molecular weights, we demonstrated the ability to tune the performance of the implant (e.g., release rates of APIs, implant duration) via PCL properties. 

#### 2.1.2. PCL MW Blends

To further tune the properties of the implant, we fabricated extruded tubes comprising binary blends of pure PCL of differing MWs (i.e., PC-08/PC-12, PC-12/PC-17, PC-08/PC-17). For each binary blend, two PCL types each with a unique MW were combined at different weight ratios. For example, PC-12 was blended with PC-17 in three different weight ratios of PC-12 to PC-17 (25:75, 50:50, and 75:25). The digital camera images of extruded tubes with PCL MW blends are shown in Appendix A. For PCL tubes with a single MW and the same wall thickness, extruded tubes fabricated with PC-08 appear most opaque, whereas the PC-17 tubes are the most translucent. This observation can be explained by the crystallinity of PCL. With similar crystallite sizes measured across three non-blended PCL types, extruded tubes with higher % crystallinity contain more crystalline domains that block the transmission of the light, resulting in higher opacity of the tubes. Interestingly, the appearance of the extruded tubes with PCL MW blends exhibits a similar trend, where the translucency of the tubes increases with an increasing weight percentage of the higher-MW PCL component within the blends (Appendix A). Extruded tubes with PCL binary blends exhibit an intermediate opacity, compared to non-blended polymer formulations. Images acquired with an optical microscope also confirmed the differences in the opacity among these PCL tubes (Appendix A). A scanning electron microscope (SEM) was used to further characterize the surface morphology, with all the extruded tubes showing relatively smooth surfaces. Some striations were present along the length of the PCL tubes (Appendix A), which is likely a feature of the extruder die. Minor surface defects were also observed in isolated regions on the tubes (Appendix A).

To assess the effects of the PCL composition on the release kinetics of TAF from the implants, we performed in vitro studies using extruded tubes with various PCL binary blends. All implants contained the same drug formulation (2:1 mass ratio of TAF to sesame oil) and had the same dimensions (wall thickness of 100 µm, 2.5 OD, 40 mm in length). The cumulative release profiles of these implants exhibited zero-order release kinetics for >4 months (Figure 4) and the estimated duration of these implants is >1 year based on the initial payload (~120 mg/implant). The cumulative percentage drug release profiles of these implants are shown in Appendix A. As expected, the release rate of TAF is a function of the PCL composition, where the release rates of TAF increase with an increasing weight percentage of the higher-MW PCL component (Table 3). For instance, implants fabricated with blends of PC-12/PC-17 showed a higher release rate of TAF with increasing content of PC-17, (i.e., 25/75 > 50/50 > 75/25 for PC-12/PC-17 tubes). Of note, implants fabricated from these three PCL binary blends of PC-12/PC-17 release TAF at rates intermediate to the release rates of implants fabricated from non-blended, pure PCL, which aligns with observations of other polymer MW blends [18]. With implants of the same blending ratio of 25/75, the release rates of TAF were affected by the MW as well: PC-12/PC-17 > PC-08/PC-17 > PC-08/PC-12. Therefore, these data indicate that the release rates of TAF are readily controlled via the MW of the polymer blends. The purity of TAF within the sesame oil formulation was evaluated at the conclusion of the study (Table 3). All the formulations showed TAF at stability of >91% after in vitro exposure of >4 months. The observed degradation of TAF is likely due to its inherent hydrolytic instability, which we previously showed degrades through non-linear profiles within our implant under simulated physiological conditions [8]. Additionally, the chromatographic purity and release rates of TAF are inversely correlated, which is in agreement with previous observations and indicates that the rate of TAF degradation is affected by the composition of PCL.

Since the distinct properties of each PCL blend formulation can affect the diffusion kinetics of TAF from the implants, we characterized each PCL MW blend using GPC, DSC, and XRD (Figure 5). As expected, the MW of each PCL formulation varies and is dictated by the composition of the blend. Figure 5 shows that the overall MW of the blended polymer formulations trends downward with decreasing quantities of higher-MW PCL within each blended formulation. For example, the overall MW of PC-17/PC-12 blends decreases as the quantity of PC-17 decreases within the formulation. The MW of PCL binary blends is intermediate to the MW of the non-blended, pure PCL (Figure 5a). For the % crystallinity as measured by DSC, a clear upward trend was also observed, where the degree of crystallinity increases as the percentage of lower-MW PCL within the blends increases (Figure 5b). Similarly, the % crystallinity of PCL blends is intermediate of non-blended PCLs. In addition, we also evaluated the effect of gamma irradiation on the properties of PCL blends. Similar to non-blended PCLs, a slight decrease in MW after the gamma irradiation was observed, whereas the sterilization process minimally affected the % crystallinity of PCL MW blends. Taken together, these results demonstrate that hot-melt blending of PCL with different starting MWs offers an effective approach for tuning the MW and crystallinity of PCL.

To explore the effect of PCL MW blending on the crystallite size, each sample was analyzed via DSC and XRD. Irrespective of the MW of PCL or the corresponding blend formulation, the DSC thermograms of extruded tubes show a melting endotherm near 60 °C. As shown in Table 4, the DSC results demonstrate a crystallite size of ~27 nm for all PCL binary blends, which are in good agreement with non-blended PCL. Similarly, the XRD diffractograms of extruded tubes showed two intense Bragg peaks at 2θ near 21.3° and 23.7°, which remained unchanged after blending pure PCL of two different MWs. The XRD analysis shows that the crystallite size ranges from ~19 to ~20 nm (see Table 4), which is consistent with that of non-blended PCL. Overall, the crystallite sizes remain consistent across all the PCL blends, and it is unlikely that crystal size alone was responsible for the differences in the release rates of TAF. Moreover, no significant changes in the crystallite sizes of the PCL were observed after exposure to gamma irradiation as a sterilization process. 

Taken together, Figure 6 illustrates the relationships between the release rate of TAF from implants comprising different PCL formulations and the MW and crystallinity of the accompanying PCL tubes without drug. These data demonstrate that blending PCL polymers of different starting MWs offers an effective approach for tuning the physical and chemical properties of PCL that readily influence the release rate of TAF from the implant. For example, PCL designs with the lowest MW typically showed both the highest degree of crystallinity and the lowest release rates of drug. This is partly explained by the % crystallinity of the PCL: as the MW of a PCL formulation decreases, the material contains a greater abundance of crystalline regions that impede drug diffusion and consequently lower the release rate of TAF. Irrespective of the MW of the PCL formulation, however, the crystallite sizes were similar and minimally influence the drug release rates. Implants formulated using PCL blends showed intermediate release profiles relative to the pure PCL types. Thus, a strong correlation exists between the release rate of the API from the implant and polymer properties. By adjusting the composition of PCL, we can further tailor the performance of the implant including the rate and duration of drug release.

### 2.2. Effect of PCL MW Blends on the Mechanical Properties of Extruded Tubes

We further explored the effects of polymer blending on the mechanical properties of extruded PCL tubes (Figure 7). Regardless of exposure to gamma irradiation or polymer formulation, all PCL tubes exhibited mechanical behavior characteristics of a flexible polymer. As shown in Figure 7, the ultimate tensile strength (UTS) for extruded tubes comprising PC-17, PC-12, and PC-08 post gamma irradiation was 41.5 ± 2.6, 35.2 ± 1.8, and 16.1 ± 0.2 MPa, respectively. Among the three types of non-blended PCL materials, tubes fabricated from PC-17 showed the highest UTS, indicating that PC-17 can withstand the highest strains being pulled or stretched before failing. As expected, the higher-MW polymer blends are more mechanically robust and typically exhibited a higher UTS compared to the lower-MW counterparts, likely due to the entanglement of longer PCL chains. For example, with extruded tubes of the same blending ratio of 75/25, the UTS scaled as PC-12/PC-17 > PC-08/PC-17 > PC-08/PC-12. This result is in good agreement with previous work showing that an increase in the MW of polypropylene (PP) profoundly improves the ductility of the PP–calcium carbonate composites [38]. Interestingly, PC-12 showed the highest percentage of elongation at breakage and the lowest elastic modulus, whereas PC-08 exhibited the lowest percentage of elongation at breakage and the highest elastic modulus. This indicates that PC-08 is the most brittle among the three unblended PCL types and easily deforms, which is likely attributed to the high crystallinity of PC-08. It has been reported that the plastic deformation behavior of semicrystalline polymer materials is controlled by the properties of both crystalline and amorphous phases [39]. As a pseudoductile polymer, PCL tends to fail by shear yielding accompanying the deformation of the crystallites. Above the Tg, the elastic modulus increases with increasing crystallinity of the polymeric matrix [40], thus PC-08 demonstrated a relatively high modulus but was still more brittle than its low crystallinity counterparts (i.e., PC-12, PC-17). This also partly explains the elasticity of the PCL binary blends. Among PC-08/PC-12 binary blends, the 25/75 blends showed the highest % elongation at breakage and the lowest elastic modulus, and the pliability of the blends decreases as the fraction of PC-12 decreases within the PCL formulation. The differences in mechanical properties among PC-08/PC-17 or PC-12/PC-17 blends at different ratios are less pronounced. In addition, gamma irradiation minimally affected the mechanical properties of the tubes. Overall, we demonstrate that the mechanical properties of the extruded tubes depend on the composition of the PCL formulation, and the mechanical properties of a polymer MW blend, such as UTS or elastic modulus, are often intermediate between those of the blend components. Therefore, we demonstrate that the mechanical properties of PCL tubes are tunable by adjusting the composition of the PCL blends and the mechanical integrity of implants is, therefore, readily enhanced via polymer blending.

To summarize, this manuscript describes unique PCL formulations for tailoring the performance of biodegradable implants for HIV prevention. Extruded tubes were prepared via hot-melt, single screw extrusion using binary blends of medical-grade PCL with different MWs (PC-08: MW of 51kDa, PC-12: MW of 72kDa, PC-17: MW of 106 kDa). The PCL binary blends with combinations of PC-08/PC-12, PC-12/PC-17, and PC-08/PC-17 were prepared in three different ratios: 25/75, 50/50, 75/25 wt.%. We detailed the thermal, physical, and mechanical properties of these PCL formulations and evaluated the performance of TAF as a model drug for implants fabricated with extruded tubes of these PCL types. By adjusting the MW of the PCL formulations, we can control the performance parameters of the implant, such as the release rate of the drug, the mechanical properties, and the biodegradation timeframe.

## 3. Materials and Methods

### 3.1. Implant Fabrication

The medical-grade PCL pellets were procured from Corbion (Amsterdam, The Netherlands) at three different weight average molecular weight (MWs): (1) average MW = 51 kDa, PURASORB PC-08, PD of 2.4; (2) average MW = 72 kDa, PURASORB PC-12, PD of 2.6; and (3) average MW = 106 kDa, PURASORB PC-17, PD of 2.6. PCL tubes were fabricated via a hot-melt, single screw extrusion process using solid PCL pellets at GenX Medical (Chattanooga, TN, USA). Before the extrusion process, all the PCL pellets were dried in a compressed air dryer at 60 °C for 4 h. To fabricate extruded tubes comprising PCL MW blends, the blending was carried out by the extrusion screw in line with a rotor speed of 5–10 rpm at a temperature ranging from 60 to 116 °C depending on the MW of the PCL. The ratio of binary PCL formulated for melt blending was 25/75, 50/50, 75/25 by weight percentage. All tubes measured 2.5 mm in outer diameter (OD) and 100 µm in wall thicknesses, as determined using a 3-axis laser measurement system and light microscopy at GenX Medical.

To fabricate the implant, PCL extruded tubes were first marked and trimmed to ~46 mm to accommodate an implant with a 40 mm reservoir length with 3 mm of headspace at both ends for sealing. The initial injection seal was created by enclosing the PCL tubes at one end using heat sealing, as previously described [28]. Empty extruded tubes with first seals were weighed prior to loading. The drug formulation was generated by first mixing pharmaceutical grade, Super Refined^TM^ Sesame Oil (Croda, Cat# SR40294, Snaith, UK) and TAF, which was graciously provided by Gilead Sciences (Foster City, CA, USA). The TAF sesame oil mixture at a 2:1 mass ratio was then ground with a mortar and pestle to create a smooth and uniform paste. The paste was loaded into the empty PCL tube using a 1 mL syringe fitted with a 14-gauge blunt tip needle. After loading the formulation to the 40 mm mark, the interior tube wall of the headspace was cleaned with a rod to remove residual drug formulations that could affect the integrity of the final seals. The filled implant was then weighed again to determine the total payload prior to creating the final seal in a similar manner to the first seal. After fabrication, all implants were photographed with a ruler to record the final dimensions. Paste area was measured with ImageJ (Version 1.50e, NIH, Bethesda, MD, USA) and release rates were normalized to the surface area of a full-sized implant (2.5 mm OD, 40 mm in length), 314 mm^2^. The end of the implants (i.e., end seals) were not included in calculations of the implant surface area.

### 3.2. Implant Sterilization

All implants were fabricated in a biosafety cabinet under aseptic conditions. After fabrication, implants were sterilized by gamma irradiation with a dose range of 18-24 kGy at Steris (Mentor, OH, USA). Implants were first packed in amber glass vials and then exposed to a Cobalt-60 gamma-ray source (Nordion Inc., Ottawa, ON, Canada) on a continuous path for 8 h.

### 3.3. In Vitro Drug Release Studies

In vitro drug release studies involved incubation of the implant in plastic tubes filled with 40 mL 1X phosphate buffered saline (PBS) (pH 7.4) at 37 °C in an incubator shaker at 100 rpm. TAF species in the release buffer was measured by ultraviolet-visible (UV) spectroscopy at 260 nm using the Synergy MX multi-mode plate reader (BioTek Instruments, Inc, Winooski, VT, USA). The solubility of TAF in 1X PBS buffer at 37 °C was determined to be 4.9 mg/mL by measuring the TAF concentration in the supernatant of a solution with excess drug powder using UV-Vis. Release buffer volume and time intervals for device transfer were chosen to fully submerge the device and to maintain sink conditions in PBS (<0.49 mg/mL). The implants were transferred to 40 mL of fresh PBS buffer twice per week in a biosafety hood under aseptic conditions. The quantity of TAF released into the PBS buffer during the time interval was calculated and the cumulative mass of drug release as a function of time was determined. The release profiles of TAF were normalized to the surface area of a full-length implant (40 mm) by dividing the release rate by the surface area of 314 mm^2^.

### 3.4. Stability Analysis of TAF Formulation

The stability of TAF formulations within the device reservoir was assessed by opening up the implant with the remaining TAF formulation, dissolving the entire reservoir contents into an organic solvent, and measuring TAF chromatographic purity using ultra-performance liquid chromatography coupled with UV spectroscopy (UPLC/UV). The analysis was performed using a Waters BEH C18 column (2.1 mm × 50 mm, 1.7 μm) under gradient, reversed-phase conditions with detection at 260 nm. One single aliquot was prepared for each implant and quantitated by linear regression analysis against a 5-point calibration curve. The purity of TAF was calculated as the % peak area associated with TAF relative to the total peak area of TAF-related degradation products (detected above the limit of detection (LOD) ≥ 0.05%). The TAF formulations within the implant were analyzed after exposure of the implant to a simulated physiological condition (i.e., 1X PBS, pH 7.4 at 37 °C) for over 4 months.

### 3.5. Characterization of PCL Extruded Tubes

#### 3.5.1. Differential Scanning Calorimetry (DSC)

Modulated differential scanning calorimetry (MDSC) (TA Instruments Q200, RCS90 cooling system, New Castle, DE, USA) was used to evaluate the melting behavior of PCL extruded tubes comprising pure PCL and PCL blends. Samples of approximately 8 mg polymer tubing were placed in a Tzero^TM^ Pan and sealed with a Tzero^TM^ Lid and a dome-shaped die, resulting in a crimped seal. After placing in a nitrogen-purged DSC cell, the sample was immediately cooled to 0 °C, then heated to 120 °C at a rate of 1 °C/min with an underlying heat-only modulation temperature scan of ±0.13 °C every 60 s. The peak temperature of the melting endotherm was taken as the melting temperature (*T_m_*) of the polymer. The peak area (between 25 and 65 °C) was used to determine the melting enthalpy. All the DSC data analysis was conducted using the TA Universal Analysis software (version 4.5A, TA Instruments, New Castle, DE, USA). The mass % crystallinity was calculated using Equation (1), where *X_c_* represents the mass fraction of crystalline domains in PCL, Δ*H_m_* represents the enthalpy of melting measured by the DSC, and Δ*H_fus_* represents the theoretical enthalpy of melting for 100% crystalline PCL, reported as 139.5 J/g [41,42].
(1)$$X_{C}=\frac{\Delta H_{m}}{\Delta H_{fus}}\times 100$$

The peak melting temperatures of polymers were used to calculate crystallite sizes within the sample using the Thompson–Gibbs equation (2) [43]:(2)$$L=\frac{2\sigma_{e}T_{m}^{o}}{\Delta H_{m}^{o}\left(T_{m}^{o}-T_{m}\right)}$$
where *L* is the crystallite size in nm, *σ_e_* is the free energy of chain folds in mJ/m^2^, *T_m_^o^* is the equilibrium melting temperature in K, *T_m_* is the melting temperature measured by DSC in K, and Δ*H_m_^o^* is the enthalpy of fusion for 100% crystalline polymer in J/g. *T_m_^o^* and Δ*H_m_^o^* were taken from the ATHAS data bank as 342.2 K and 139.5 J/g, respectively. The free energy associated with chain folding was taken as 60 mJ/m^2^ [44].

#### 3.5.2. X-ray Diffraction (XRD)

The X-ray diffractograms of the extruded PCL tubes comprising non-blended or binary blends at a wall thickness of 100 µm were acquired using a Bruker AXS, Inc. D8 Advance model utilizing standard Bragg–Brentano geometry and a LynxEye XE-T high-resolution detector. The extruded polymer tubing was first cryo-grinded in a freezer mill using liquid nitrogen for 1.5 min after cooling for 3 min before initiating the grinding cycle. Samples were packed into a zero background sample holder and scanned at a voltage of 40 kV and a current of 40 mA (1600 Watts). The scanning angle ranged from 5° to 70°, with a step size of 0.02° and a dwell time of 2 s per step. The XRD data were analyzed using the MDI Jade version 9.6 software and the 2019 ICDD PDF 4+ database was used to search match crystalline phases present in the materials. The crystallite size was determined via the Scherrer equation (3): (3)$$L=\frac{K\lambda }{\beta cos\theta }$$
where *L* = crystallite size, *K* = Scherrer constant (0.94 from literature) [45,46], *λ* = X-ray wavelength, *β* = full-width at half maximum of a crystallographic peak, and *θ* = Bragg angle.

#### 3.5.3. Gel Permeation Chromatography (GPC)

Approximately 10 mg of extruded polymer tubing comprising pure PCL or PCL MW blends was dissolved in 1 mL of tetrahydrofuran (THF). Then, 40 µL of the solution was injected into an Agilent 1100/1200 HPLC-UV instrument (Santa Clara, CA, USA, flow rate of 1.0 mL/min). The mobile phase was THF with a flow rate of 1.0 mL/min. The weight average molecular weights (MW) and polydispersity (PD) were determined. The calibration was done prior to sample analysis using polystyrene polymer standards (MWs of 2460 to 0.545 kDa).

#### 3.5.4. Scanning Electron Microscopy (SEM) and Optical Microscope

Surface features of the extruded tubes comprising PCL homopolymers and binary blends were visualized by scanning electron microscopy (Quanta 200; FEI, Hillsboro, OR, USA) at an accelerating voltage of 20 kV and a spot size of 2.5 under high vacuum at various magnifications. Samples were prepared for imaging by slicing the extruded tubes and flattening them onto a carbon adhesive substrate mounted to an aluminum sample holder sputter coated with Au/Pt (Hummer Sputtering System, Anatech Ltd., Union City, CA, USA) under argon for 120 s. The surface morphology of the PCL extruded tubes was also characterized using an Olympus SZ61 stereomicroscope (zoom range 0.67× to 4.5×). The microscope is equipped with an Olympus SC30 camera that allowed sample photography. Similarly, the extruded tubes were sliced open and flattened to adhere to glass slides.

#### 3.5.5. Tensile Testing

The tensile tests were performed using a MTS Electromechanical testing system with a calibrated load-cell of 1250 N. Specimens (60 mm length, 2.5 mm OD, and 2.3 mm inner diameter) were tested in 5–10 replicates at room temperature by stretching the sample at a constant deformation rate of 200 mm/min. The original data were analyzed by the software of TestWorks^®^ (MTS System Corporation). The resulting stress–strain data were used to calculate the elastic modulus, % elongation, and UTS. The elastic modulus was defined as the slope of the linear portion of the stress-strain curve, which occurred in the range of 0–100% of the UTS. % elongation at the breakage is defined as the maximum elongation of the sample length divided by the original sample length. The peak stress achieved during mechanical testing was taken as the UTS.

## 4. Conclusions

This manuscript describes new PCL formulations for a biodegradable subcutaneous implant for HIV PrEP. Overall, the choice of PCL formulation is critical to meet the targeted product specifications of the implant platform. This paper shows the utility of new PCL formulations to meet targeted drug release rates and mechanical properties of a biodegradable long-acting implant for HIV PrEP.

## Figures and Tables

**Figure 1 ijms-22-06529-f001:**
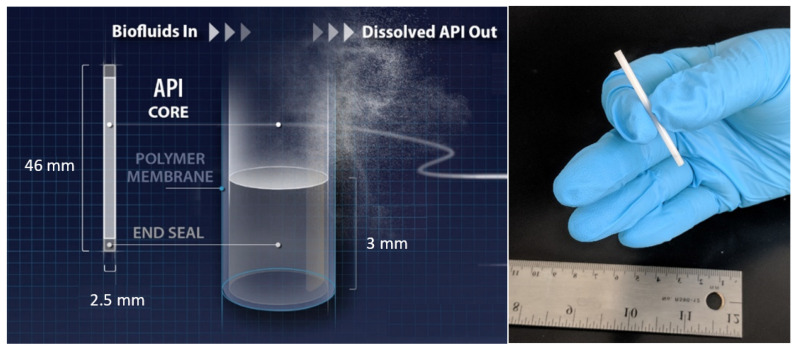
(**Left**) A schematic of a PCL reservoir-style implant for delivery of TAF. All implants were 2.5 mm in outer diameter (OD) and 46 mm in total implant length with a 40 mm paste and two 3 mm PCL end seals. (**Right**) A digital camera image of the biodegradable implant.

**Figure 2 ijms-22-06529-f002:**
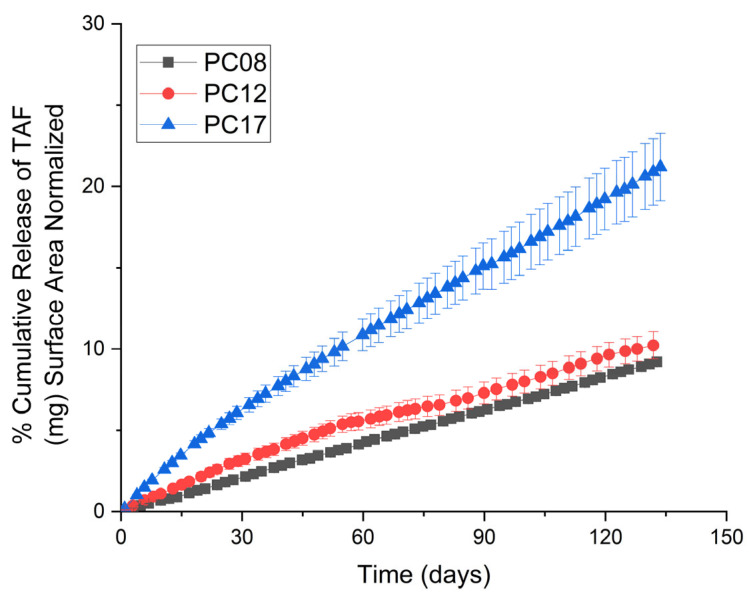
Cumulative release profiles of TAF from implants comprising extruded tubes of PC-08, PC-12, and PC-17. All implants contain a formulation of 2:1 TAF/sesame oil and tubes with a wall thickness of 100 µm, a length of 40 mm, and an OD of 2.5 mm. All samples were performed in triplicate.

**Figure 3 ijms-22-06529-f003:**
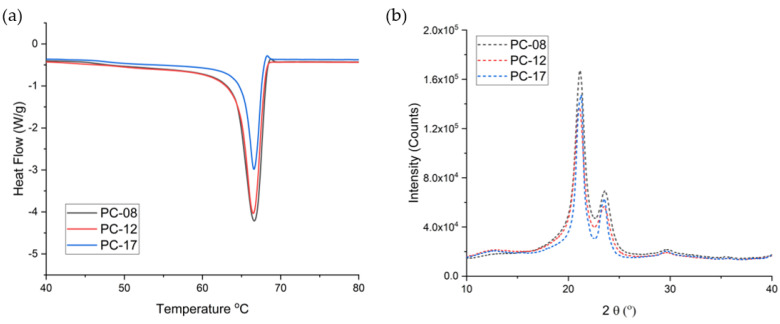
Exemplary graphs of (**a**) DSC heat flow curves and (**b**) XRD profiles of PCL tubes comprising PC-17, PC-12, and PC-08 with 100 µm wall thickness and 2.5 mm OD.

**Figure 4 ijms-22-06529-f004:**
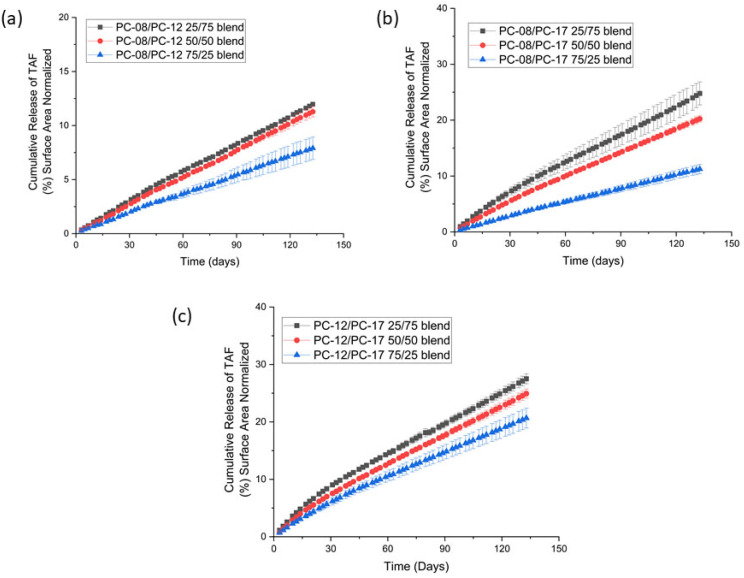
Cumulative release profiles of TAF from implants fabricated with PCL MW blends comprising (a) PC-08/PC-12 binary blends, (b) PC-08/PC17 binary blends, and (c) PC-12/PC-17 binary blends. All implants were formulated with a 2:1 mass ratio of TAF to sesame oil, a wall thickness of 100 µm, a length of 40 mm, and OD of 2.5 mm. Implants were gamma-irradiated before initiating the in vitro study. All samples were performed in triplicate.

**Figure 5 ijms-22-06529-f005:**
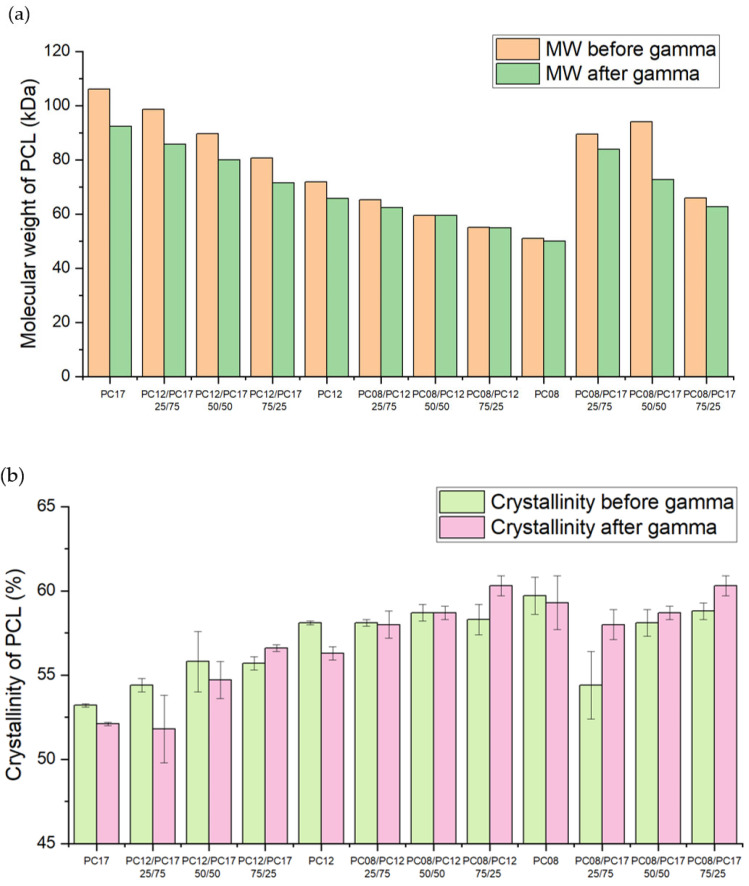
(**a**) The weight average MW and (**b**) the crystallinity of the extruded tubes comprising PCL binary blends after gamma irradiation. The extruded tubes had a wall thickness of 100 µm and an OD of 2.5 mm. The % crystallinity reflects the mean values ± standard deviation of three independent measurements, whereas the MW value was reported based on one measurement.

**Figure 6 ijms-22-06529-f006:**
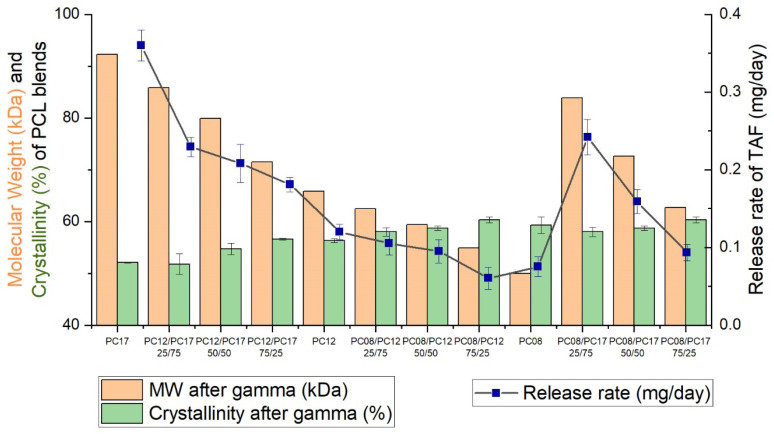
The relationships between the release rate of TAF and various properties of PCL formulations. The release rates of TAF were normalized to the surface area of an implant with dimensions of 2.5 mm OD and 40 mm length (i.e., 314 mm^2^). The in vitro release assessments were performed in triplicate. The weight average MW and % crystallinity were determined using hollow extruded tubes without drug formulations. The % crystallinity reflects the mean values ± standard deviation of three independent measurements, whereas the MW value was reported based on one measurement.

**Figure 7 ijms-22-06529-f007:**
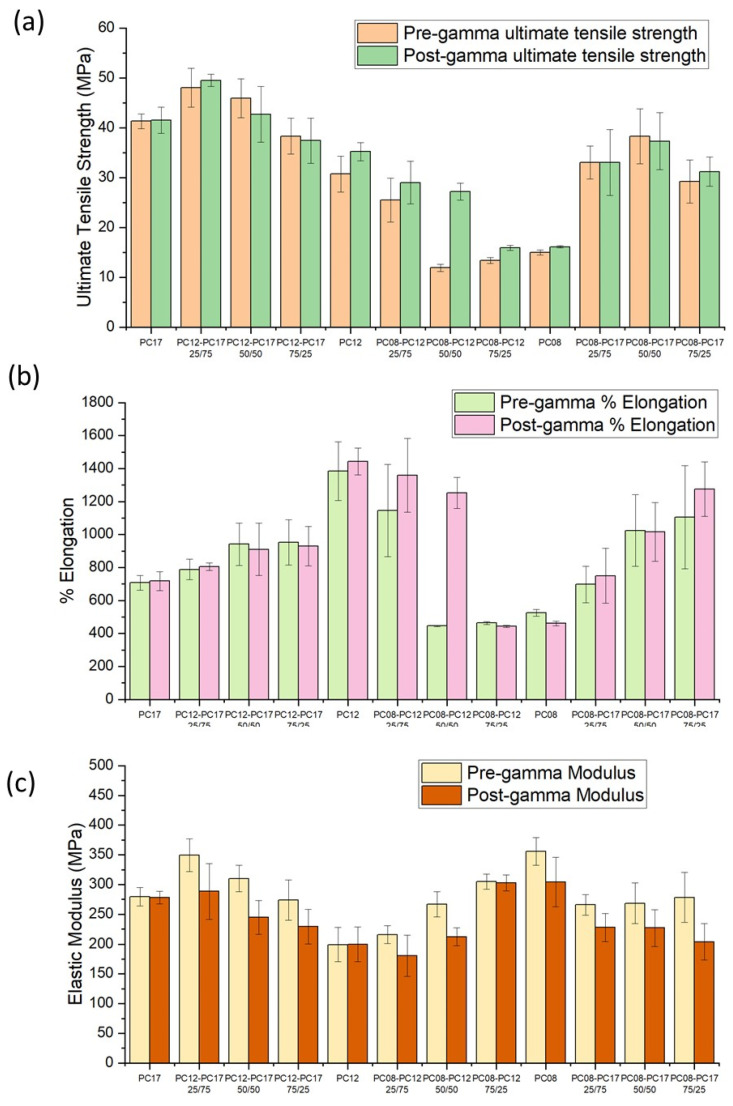
(**a**) The ultimate tensile strength, (**b**) percentage elongation at the break, and (**c**) elastic modulus of extruded Table 2. 5 mm OD, 60 mm length, and 100 µm wall thickness. The UTS, %elongation, and elastic modulus reflect the mean values ± standard deviation of at least 5 samples.

**Table 1 ijms-22-06529-t001:** The weight average MW and crystallinity of tubes extruded from different non-blended PCL types (PC-08, PC-12, PC-17) as determined by DSC, and the average release rates of TAF from implants fabricated with tubes of these PCL types. Implants contain a formulation of 2:1 TAF/sesame oil and tubes have a wall thickness of 100 µm, a length of 40 mm, and an OD of 2.5 mm. The in vitro release assessments were performed in triplicates. The MW and % crystallinity were determined using hollow extruded tubes without drug formulations. The % crystallinity reflects the mean values ± standard deviation of three independent measurements, whereas the MW value was reported based on one measurement.

PCL Type	MW (kDa) before Gamma	MW (kDa) after Gamma	Crystallinity before Gamma (%)	Crystallinity after Gamma (%)	Average Release Rate of the Implant (mg/day) *
PC-17	106.0	92.3	52.56 ± 0.62	53.19 ± 0.12	0.36 ± 0.02
PC-12	71.9	65.8	56.27 ± 0.44	58.15 ± 0.13	0.12 ± 0.01
PC-08	50.9	50.0	59.26 ± 1.93	59.69 ± 1.10	0.07 ± 0.01

* Implants were exposed to gamma irradiation before initiating the in vitro analysis.

**Table 2 ijms-22-06529-t002:** Thermal properties of PCL tubes from XRD and DSC analysis.

PCL Type	Crystallite Size before Gamma (nm)-DSC	Crystallite Size after Gamma (nm)-DSC	Crystallite Size before Gamma (nm)-XRD		Crystallite Size after Gamma (nm)-XRD	
			L_110_	L_200_	L_110_	L_200_
PC-17	27.5 ± 0.2	26.9 ± 0.2	10.7	8.9	11.7	9.9
PC-12	26.9 ± 0.1	26.2 ± 0.2	10.1	8.2	10.6	8.3
PC-08	27.3 ± 0.5	27.5 ± 1.5	10.6	8.5	10.7	8.4

Extruded tubes comprised 100 µm wall thickness and 2.5 mm OD. The DSC analysis was performed in triplicate, whereas the crystallite size measured by XRD was reported based on a single measurement.

**Table 3 ijms-22-06529-t003:** The average release rates of TAF from implants fabricated with PCL MW blends. The percent purity of TAF was determined at the conclusion of the in vitro studies (i.e., day 136). The approximate quantity of drug payload for implants comprising each PCL formulation. Implants were gamma-irradiated before initiating the in vitro study. All samples were performed in triplicate.

Formulation	Approximate TAF Payload (mg)	Average Release Rate (mg/day, 40 mm Implant)	% Purity of TAF at Day 136
PC-12/PC-17 25/75	114 ± 5	0.33 ± 0.01	91.2 ± 3.1
PC-12/PC-17 50/50	116 ± 2	0.23 ± 0.04	96.2 ± 0.8
PC-12/PC-17 75/25	116 ± 4	0.18 ± 0.01	94.8 ± 0.7
PC-08/PC-12 25/75	113 ± 2	0.10 ± 0.01	96.9 ± 0.1
PC-08/PC-12 50/50	112 ± 2	0.09 ± 0.01	96.4 ± 0.4
PC-08/PC-12 75/25	112 ± 2	0.06 ± 0.01	96.7 ± 0.1
PC-08/PC-17 25/75	115 ± 4	0.24 ± 0.02	95.2 ± 0.4
PC-08/PC-17 50/50	108 ± 3	0.16 ± 0.02	93.3 ± 1.1
PC-08/PC-17 75/25	119 ± 3	0.09 ± 0.01	95.1 ± 0.3

**Table 4 ijms-22-06529-t004:** Thermal properties of PCL tubes comprising binary blends from XRD and DSC analysis. The extruded tubes had a wall thickness of 100 µm and an OD of 2.5 mm. The DSC analysis was performed in triplicates, whereas the crystallite size measured by XRD was reported based on a single measurement.

PCL Type	Crystallite Size before Gamma (nm)-DSC	Crystallite Size after Gamma (nm)-DSC	CRYSTALLITE Size before Gamma (nm)-XRD		Crystallite Size after Gamma (nm)-XRD	
			**L_110_**	**L_200_**	**L_110_**	**L_200_**
PC-08/PC-12 25/75	26.6 ± 0.3	26.2 ± 0.2	10.4	10.3	11.1	10.6
PC-08/PC-12 50/50	26.5 ± 0.2	27.0 ± 0.1	10.8	8.6	10.6	8.3
PC-08/PC-12 75/25	27.0 ± 0.2	26.8 ± 0.1	11.2	10.6	11.0	8.3
PC-12/PC-17 25/75	27.3 ± 0.1	26.9 ± 0.2	11.0	8.9	10.6	8.5
PC-12/PC-17 50/50	27.2 ± 0.2	27.2 ± 0.8	10.6	10.4	10.8	8.5
PC-12/PC-17 75/25	26.9 ± 0.2	27.8 ± 0.3	11.0	8.6	11.0	8.7
PC-08/PC-17 25/75	27.3 ± 0.4	27.3 ± 0.2	10.6	10.3	11.0	10.6
PC-08/PC-17 50/50	27.2 ± 0.4	26.9 ± 0.2	11.0	10.4	10.4	8.1
PC-08/PC-17 75/25	26.9 ± 0.2	26.8 ± 0.4	10.7	8.5	11.1	8.6

## Data Availability

The data presented in this study are available on request from the corresponding author.

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
