# Peer review of "Effects of Polymer Blending on the Performance of a Subcutaneous Biodegradable Implant for HIV Pre-Exposure Prophylaxis (PrEP)"

_ijms, 2021, doi:10.3390/ijms22126529_

Round 1

Reviewer 1 Report

Johnson and coworkers reported the formulation of PCL-implant for sustained release of tenofovir alafenamide. Several following comments shall be addressed:

  1. Line 103, please explain the mechanism of encapsulation, i.e. the interaction between the implant and the drug.
  2. Line 116, the dispersity of each polymer shall be provided.
  3. What is the loading efficiency and encapsulation efficiency of these formulations listed in Table 3 for the model drug?
  4. Several relevant references should be included: DOI: 10.1021/bm5002262; DOI: 10.1021/acs.biomac.5b01020; DOI: 10.1021/acs.macromol.9b02595; DOI: 10.1021/acs.macromol.7b01918.  

Reviewer 2 Report

Linying Li et al investigated the effect of polymer blending on the performance including in vitro release rates, crystallinity, and mechanical properties of biodegradable implants for the use of HIV PrEP.

  1. The right side image of figure 1 is already published in the previous paper (reference 24). Replace it with a different image for depicting the implant.
  2. There is no information on the number of replicates and mean±SD/SEM in the figures and tables. The authors should include the mean±SD/SEM and number of replicates information for all figures and tables.
  3. Figure 3 images are not clear and data legends are not distinguishable. The authors should provide clear images with distinguished data legends.
  4. What are the % drug loading, drug content, % water content, content uniformity, and % cumulative release rate of the implants? The authors should include all this information in the manuscript.
  5. The authors need to include the detailed procedure of the in vitro release study including rpm, aliquot volumes, sink conditions, etc.
  6. The conclusion is long and should explain by supporting the overall results.
  7. The authors should include a detailed description in the supplementary figure legends like magnification used for the microscopic images.
  8. This manuscript has similar content at several places from the previously published papers (reference 23 and 24). The authors should thoroughly rewrite the manuscript to minimize the plagiarism content.

Round 2

Reviewer 2 Report

Linying Li et al made significant improvements to the manuscript and answered most of the comments.

Figure S10 images are not clear. The authors need to incorporate clear images with high resolution.

Why the authors haven’t performed the in vivo studies? The authors should explain how the implant formulations will achieve sustained and target concentrations in in vivo animal models and clinical studies for the prevention of HIV PrEP.

Author Response

Reviewer #2

Linying Li et al made significant improvements to the manuscript and answered most of the comments.

We thank this reviewer for his or her kind comments and positive assessment of our revised work. We appreciate all the thoughtful feedback and constructive suggestions.

  1. Figure S10 images are not clear. The authors need to incorporate clear images with high resolution.

[See attached files.]

We have updated Figure S10 with images of better resolution as below and in the revised Supplementary Information.

Figure S10. Percentage Cumulative release profiles of TAF from implants comprising extruded tubes of PCL MW blends. All implants contain a formulation of 2:1 TAF/ sesame oil and tubes with a wall thickness of 100 µm, a length of 40 mm, and an OD of 2.5 mm. All samples were performed in triplicate.

  1. Why the authors haven’t performed the in vivo studies? The authors should explain how the implant formulations will achieve sustained and target concentrations in in vivo animal models and clinical studies for the prevention of HIV PrEP.

We have evaluated our TAF implants in several in-vivo studies that involve New Zealand rabbit1, beagle dog2, and rhesus macaque3 models. We demonstrate that the TAF implants delivered sustained plasma TAF/TFV concentrations and maintained PBMC Tenofovir diphosphate (TFV-DP) levels at putative protective concentrations for up to 6 months in beagle dogs. For this dog study, these implants were effectively inserted with a Sino-implant trocar and easily retrieved from at scheduled time points. Besides evaluating the pharmacokinetic (PK)  profiles of the implants, we also conducted an efficacy study in pigtail macaques where we demonstrate TAF implants releasing 0.7 mg/day resulted in high TFV-DP levels in PBMCs that provided complete protection against vaginal Simian-Human Immunodeficiency Virus (SHIV) infection.3 In this current manuscript, the in-vitro release rates of TAF formulations are tailored to range between ~0.1 mg/day to ~0.4 mg/day from implants comprised of various PCL formulations. Previously, we have demonstrated a release rate of 0.8 mg/day of TAF from a single device at 45 µm wall thickness (Ref 28). Thus, we will be able to achieve the target dosing via a single thinner-walled implant or multiple implants with a thicker wall. Additionally, using multiple implants to achieve the desired dosing could extend the therapeutic duration to longer periods with protection, which is evidenced by Probuphine® (4 rods) and Norplant® (6 rods).

References:

  1. Development of an End-User Informed Tenofovir Alafenamide (TAF) Implant for Long-acting (LA)-HIV Pre-exposure Prophylaxis (PrEP), GJ Gatto, RM Brand, N Girouard, L Li, L Johnson, MA Marzinke, E Krogstad, A Siegel, Z Demkovich, E Luecke, A van der Straten, HIV R4P, 2018
  2. Sustained 6-month Release of Tenofovir Alafenamide (TAF) From A Biodegradable Implant For Long-acting (LA)-HIV Pre-Exposure Prophylaxis (PrEP), GJ. Gatto, SA Krovi, L Johnson, ZR Demkovich, MA Marzinke, E Luecke, A van der Straten, CRS, 2020
  3. High protection against vaginal SHIV infection in macaques by a biodegradable implant releasing tenofovir alafenamide. I. Massud, K. Nishiura, S. Ruone, A. Krovi, A. Holder, J. Gary, P. Mills, J. Mitchell, G. Khalil, L. Li, L. Johnson, E. Luecke, W. Heneine, G. Garcá½·a-Lerma, C. Dobard, A van der Straten, R4P, 2020
